# Forecasting the dissemination of antibiotic resistance genes across bacterial genomes

Mostafa M. H. Ellabaan [1 ✉], Christian Munck[1], Andreas Porse[1], Lejla Imamovic[1] & Morten O. A. Sommer [1 ✉]

Antibiotic resistance spreads among bacteria through horizontal transfer of antibiotic resistance genes (ARGs). Here, we set out to determine predictive features of ARG transfer among bacterial clades. We use a statistical framework to identify putative horizontally transferred ARGs and the groups of bacteria that disseminate them. We identify 152 gene exchange networks containing 22,963 bacterial genomes. Analysis of ARG-surrounding sequences identify genes encoding putative mobilisation elements such as transposases and integrases that may be involved in gene transfer between genomes. Certain ARGs appear to be frequently mobilised by different mobile genetic elements. We characterise the phylogenetic reach of these mobilisation elements to predict the potential future dissemination of known ARGs. Using a separate database with 472,798 genomes from Streptococcaceae, Staphylococcaceae and Enterobacteriaceae, we confirm 34 of 94 predicted mobilisations. We explore transfer barriers beyond mobilisation and show experimentally that physiological constraints of the host can explain why specific genes are largely confined to Gram-negative bacteria although their mobile elements support dissemination to Gram-positive bacteria. Our approach may potentially enable better risk assessment of future resistance gene dissemination.

[1] The Novo Nordisk Foundation Center for Biosustainability, Technical University of Denmark, Lyngby, Denmark. ✉email: mostafa.mhashim@gmail.com; msom@bio.dtu.dk

Antibiotic resistance can evolve through horizontal acquisition of antibiotic resistance genes (ARGs)[1]. Horizontal gene transfer (HGT) has led to the evolution of resistant pathogens such as methicillin-resistant *Staphylococcus aureus*, extended spectrum β-lactamase-producing Enterobacteria, and vancomycin-resistant Enterococci[2]. These pathogens received a resistance gene from another bacterial species, linking clinically important pathogens to the resistome, a global reservoir of resistance genes that confer antibiotic resistance when expressed in a sensitive host[3–7].

ARGs transition from environmental reservoirs to human pathogens in a multistage process of initial mobilisation followed by one or more dissemination and adaptation events. Mobilisation and dissemination processes have been investigated for specific ARGs such as the *cmx* genes[7], the *ctx-m* genes[8], and the vancomycin-resistance operon[9].

Mobile genetic elements (MGEs) such as transposons and genes that encode enzymes that facilitate them such as integrases or recombinases often facilitate the initial mobilisation. Such MGEs are capable of capturing ARGs from chromosomes and horizontally transferring them via a plasmid or a phage to other bacteria[7–11]. Several studies have used complete genomes and metagenomic datasets to identify the key forces underlying HGT of ARGs[12–15]. Mapping recent gene transfers across ~2700 genomes found overlapping ecological habitats to be a major factor in shaping HGT among microbes[12]. Analysis of mobile ARGs and their neighbouring mobilisation elements across 23,425 genomes found that phylogeny is another major variable shaping networks for resistance gene transfer[13]. This may be driven by biochemical interaction between acquired genes and the cellular machinery making the acquired genes less likely to function across phylogenetically distant hosts[14]. These studies provide a fundamental understanding of the mechanisms and networks driving ARG dissemination including some of the barriers to gene dissemination.

While key forces underlying HGT are only starting to unfold, there is a lot to be explored with regard to gene mobilisation and dissemination. The ongoing acquisition of ARGs by human pathogens fuels interest in methods to predict dissemination of resistance genes. Yet such efforts are complicated by the vast number of genes in environmental and manmade reservoirs that can confer resistance[4,15,16]. Indeed, ARGs in human pathogens are vastly outnumbered by the quantity of genes in environmental microbiomes that can confer antibiotic resistance in pathogens. Strategies to identify candidates for future clinically relevant resistance genes include ranking ARGs based on the spectrum or clinical use of the antibiotics against, which they confer resistance or the presence of nearby mobilisation elements[17]. Predicting when antibiotic resistance is likely to emerge in a bacterial population is essential to uncovering the fundamentals of resistance transfer, and for the design preventive measures to limit the emergence of resistance. Yet, the establishment of the computational tools required for this type of prediction are yet to be available. Despite the several studies proposed to identify the gene exchange networks[12–15] of ARGs, none so far has been proposed a systemic framework to identify the future dissemination of ARGs, considering current data available on the dissemination of ARGs and thier associated MGEs.

In this work, we deploy a statistical framework to elucidate dissemination networks for ARGs and their associated mobilisation elements (Supplementary Fig. 1). Using this information, we predict the dissemination potential of currently known ARGs.

## Results

**Transferable ARGs define diverse gene exchange networks of bacteria.** To identify putative horizontally transferred ARGs, we used a statistical test based on the assumption that genes transferred horizontally between two organisms are significantly more conserved than their 16S rRNA genes[6,7,12,13]. Annotated ARGs were considered horizontally transferable if pairwise alignment distances were significantly shorter for the resistance genes than for the 16S rRNA genes of their hosts (see "Methods" section). We refer to the group of organisms that passed this statistical test for a given ARG as a gene exchange network (GEN) (Supplementary Fig. 1 and Fig. 1a). We developed a statistical gene-transfer pipeline with a comprehensive, manually curated ARG database of 1799 genes (Supplementary Data 1). To reduce redundancy in the analysis, the resistance genes were clustered at 95% identity and coverage. Using BLAST, the resistance gene database was queried against 56,716 curated genomes. We identified 152 ARGs predicted with high confidence to be horizontally transferred (Fig. 1b and Supplementary Data 2). Each predicted ARG was shared by at least two species and the combined GENs comprised a total of 895 bacterial species (Supplementary Data 3). The most widely disseminated genes, ranked by number of genera participating in their GEN, included clinically relevant genes conferring resistance to antibiotic classes such as β-lactams, sulphonamides, aminoglycosides, and tetracyclines (Fig. 1b and Table 1).

We observed transferable ARGs in 22,963 bacterial genomes representing 7 phyla and ~40% of the genomes in the database (Fig. 1c). These genomes contained a median of four resistance genes predicted with high confidence to be horizontally transferred (Fig. 1d). The identified GENs often spanned diverse phylogenies with ~52% of the GENs having species from a single phylum and ~48% having species from two or more phyla, highlighting that cross-phylum ARG dissemination is common (Fig. 1c, e, f and Supplementary Fig. 2A, B). Notably, ~38% of GENs included both Gram-positive and Gram-negative bacteria, highlighting the potential for ARG transfer across this physiological division (Fig. 1c). Proteobacteria were highly involved in resistance gene exchange with the most frequently involved genera within the Enterobacteriaceae family (Supplementary Fig. 2C, D)[18–20].

**Resistance-associated MGEs are widely disseminated across bacterial genomes.** MGEs can transfer ARGs. Using genetic markers of MGEs such as transposases, integrases, and other recombinases, we identified putative MGEs in the neighbouring genetic regions of ARGs (see "Methods" section). We excluded plasmid and phage genes that may be involved in the function of plasmids or phages but not directly contribute to mobilisation of the ARG-containing DNA segments. The main reason for excluding these genes is that identifying phages and plasmids at the gene level is difficult. Multiple genetic elements may be involved in the construction and functions of phages and plasmids. Random bacterial genome dynamics could bring those elements close to ARGs. However, these genes will not capture ARGs from the genome to a plasmid or phage. Only MGEs such as insertion elements, transposases, or other mobile genes will transfer ARGs from a genome to a phage or plasmid. We have adopted this very conservative approach to avoid false positives. We considered the resulting pool of MGEs as a small but high-confidence sample (Supplementary Data 4).

To identify horizontally transferred MGEs, we applied the GEN pipeline (Supplementary Fig. 1 and Fig. 2a) using the 1182 MGEs associated with transferable ARGs as the query database

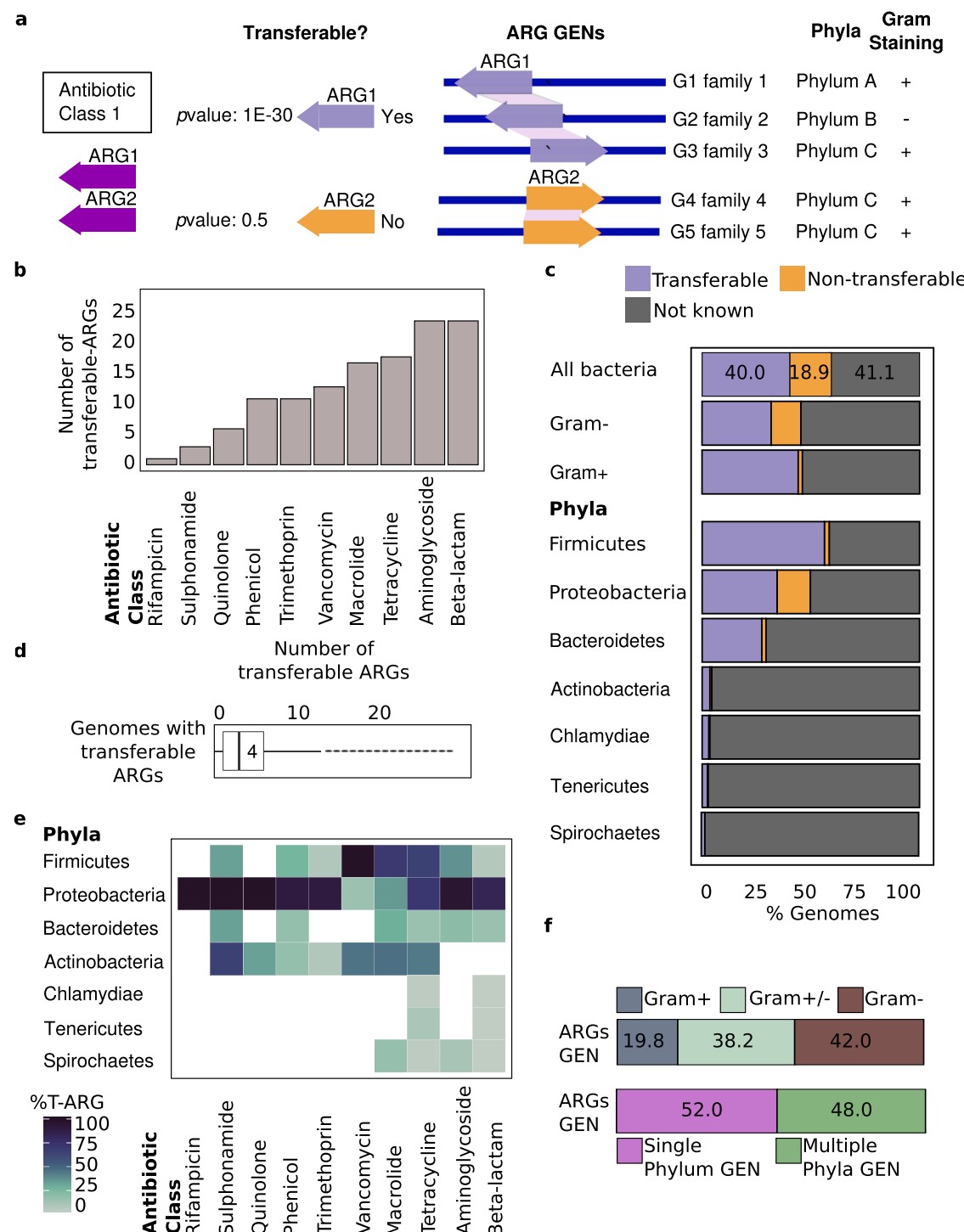

**Fig. 1 Dissemination of resistance genes summarized by antibiotic class and phylum. a** Conceptual guide linking antibiotic classes to phyla and gram staining through transferable ARGs and genomes in ARGs gene exchange networks (GENs). **b** Number of transferable ARGs per antibiotic class. **c** Percent of genomes in indicated phyla with transferable ARGs, nontransferable ARGS, or no known resistance genes. **d** Distribution of transferable ARGs across genomes. Box indicates range, line within box indicates median for all genomes, dark dots indicate outliers. **e** Heatmap of % transferable ARGs (T-ARGs) per antibiotic class observed for indicated phyla. **f** Distribution of genomes with transferable ARGs in GENs. Top, percent of GENs with Gram-negative and -positive genomes (+/− means both). Bottom, percent of GENs with one phylum or with multiple phyla. Source data are provided in Supplementary Data 1, 2, 3, and 14.

(Supplementary Data 4). Only 274 MGEs, representing 29 MGE families, were predicted with high confidence to be transferable across species (Supplementary Data 5), highlighting the conservative assumptions underlying our statistical framework. Accordingly, a majority of MGEs likely have a narrow within-species host range that cannot be resolved using our method or were not sufficiently captured in our genome dataset. Transferable MGEs associated with ARGs were found in 22,595 genomes (Supplementary Data 5), representing 39.8% of the genome dataset.

**Table 1 Highly disseminated genes based on number of genera that contain them. Source data are provided in Supplementary Data 1 and 3.**

| Antibiotic class | Gene name | ARG cluster[a] | Number of genera |
|---|---|---|---|
| β-lactam | tem-1 | Cluster960 | 59 |
| Sulphonamide | sul1 | Cluster725 | 52 |
| Aminoglycoside | aph(3″)-Ib | Cluster698 | 44 |
| Aminoglycoside | aph(6)-Id | Cluster1027 | 44 |
| Tetracycline | tetW | Cluster170 | 43 |
| Sulphonamide | sul2 | Cluster950 | 42 |
| Tetracycline | tetC | Cluster462 | 41 |
| Tetracycline | tetM | Cluster166 | 40 |
| Phenicol | catI | Cluster1371 | 37 |

ARG antibiotic resistance gene.
[a]ARGs clustered at 95% identity.

Transferred MGEs were disseminated across a median of three bacterial families (Fig. 2b). MGEs varied in their phylogenetic reach, with ~12% confined to a specific genus and ~21% able to move between different phyla (Fig. 2c). Some MGEs such as IS1 and IS240 were capable of crossing barriers between Gram-positive and Gram-negative bacteria. Other MGEs such as those belonging to IS166 may be confined to a genus such as Corynebacterium. The phylum containing the most disseminated MGEs was Proteobacteria (Fig. 2d and Supplementary Fig. 3A). Clinically relevant bacterial families with diverse MGEs were Enterobacteriaceae, Enterococcaceae, Staphylococcaceae, and Moraxellaceae (Table 2). The abundance of MGEs strongly correlated with the abundance of the transferred ARGs (Fig. 2e).

Ranking transferable MGEs based on the number of different ARGs they were associated with revealed that the most diverse MGEs belonged to the IS1, IS240, and Tn3 families, with the IS240 family displaying the broadest phylogenetic reach (Fig. 2f and Supplementary Data 6–7). Genes conferring resistance to aminoglycoside, tetracycline, or β-lactam antibiotics had the highest number of unique MGEs surrounding them, explaining the wide dissemination of these ARGs (Fig. 2g, Supplementary Fig. 3B, and Supplementary Data 8).

**Mobile genetic context predicts dissemination potential of ARGs**. When comparing the GENs of ARGs to the GENs of ARG-associated MGEs, we observed larger networks for MGEs than for their associated ARGs (Fig. 3a, b). We hypothesised that the current dissemination of MGEs could be used to predict potential future dissemination of neighbouring transferable ARGs. An ARG mobilised by an MGE may not currently be observed in all species that can host the MGE (Fig. 3a, b). Consequently, the dissemination potential of an ARG could be expected to include species in which a nearby mobilisation element has been observed. To assess the dissemination potential of currently known ARGs, we identified genomes in which the MGE was present but a neighbouring transferable ARGs had not been observed (Supplementary Data 9).

Based on this analysis, 101 (~66%) transferable ARGs had the potential to reach a new host (Fig. 3c). In total, 463 species with no observed transferable ARG could potentially receive transferable ARGs based on their existing MGE spectrum (Fig. 3b). On average, each ARG could reach an additional 164 species, 44 genera, and 21 families, indicating that the transferable resistome still has substantial dissemination potential (Supplementary Figs. 4–6). At the phylum level, 84 of transferable ARGs (more 55% of the transferable resistome) were predicted to be able to

reach a new phylum. Actinobacteria, cyanobacteria, firmicutes, and proteobacteria will likely be the most prominent future recipients of ARGs. In addition, 52 transferable ARGs not yet observed in actinobacteria, have neighbouring MGEs observed within this phylum.

An example of an ARG with high dissemination potential is ctx-m-125 (Fig. 3d). Its current GEN has three families (Yersiniaceae, Enterobacteriaceae, and Morganellaceae) and it is found associated with four transferable MGEs: IS1, IS240, Tn3 and IS110. These MGEs are currently found in 195 new species representing 31 Gram-positive and Gram-negative bacterial families, giving ctx-m-125 substantial dissemination potential (Fig. 3d). ctx-m-125 has not been observed in Pseudomonadaceae, yet three MGEs neighbouring ctx-m-125 (IS1, IS240, and Tn3) are found in Pseudomonadaceae. Accordingly, ctx-m-125 may be disseminated by these three MGEs into Pseudomonadaceae in the future unless functional or ecological factors limit its movement.

ARG groups with the highest dissemination potential are β-lactamase genes such as in the ctx-m and oxa families, tetracycline-resistance genes such as tetC, aminoglycoside-resistance genes such as aac(6')-Ia, aph(6)-Id and aadA, and macrolide-resistance genes such as ermB and mphA (Supplementary Fig. 4 and Supplementary Data 10).

Our prediction of the future dissemination of resistance genes (Fig. 4) shows potential for ARGs reaching new genera in important pathogenic families, conferring resistance to antibiotic classes currently not observed in those families (Supplementary Fig. 2A).

**Predicted dissemination confirmed in sequence read archive**. Sequence read archive (SRA) is a public repository of sequence data (of any kind, including many raw metagenomes samples). We considered only whole bacterial genome sequence data in SRA. We obtained 472,798 genomes from SRA to test our predictions of ARG dissemination, examining ARGs predicted to transfer to Enterobacteriaceae, Staphylococcaceae, and Streptococcaceae (see "Methods" section). The three families were chosen because of their deposited data and represent more than 50% of the SRA bacteria genomes (~887,000 genomes). These families represent both Gram-positive and Gram-negative bacteria, are the most sampled families, and include many common human bacterial pathogens.

We searched all sequence reads from each family in SRA and found 62,209 genomes for Staphylococcaceae, 96,376 for Streptococcaceae, and 314,213 for Enterobacteriaceae. These genome datasets represented 8.4-times to 32.9-times the number of genomes for these families in our original genome database and almost ~50% of the whole genomes available in SRA.

Predictions of future dissemination of ARGs was based on the difference between the phylogenetic reach of the ARGs and their associated MGEs, while the confirmation analysis was based on finding ARGs in the predicted hosts using the independent data set of SRA genomes (Fig. 5a). For example, the ARGs catI and tetH were not observed in Staphylococcaceae and Enterobacteriaceae in the initial GEN (Supplementary Data 3). However, GENs of their neighbouring MGEs IS1 and IS240 suggested Staphylococcaceae and Enterobacteriaceae as future hosts for catI and tetH. Our analysis of SRA genomes found MGEs IS1 and IS240 and ARGs catI and tetH within 10 kb of each other on assembled Staphylococcaceae and Enterobacteriaceae genomes (Fig. 5b, c). These observations confirmed our predictions that MGEs in the neighbourhood of ARGs can carry those genes to new families. This confirmation supports the use of our approach to predict dissemination of transferable ARGs.

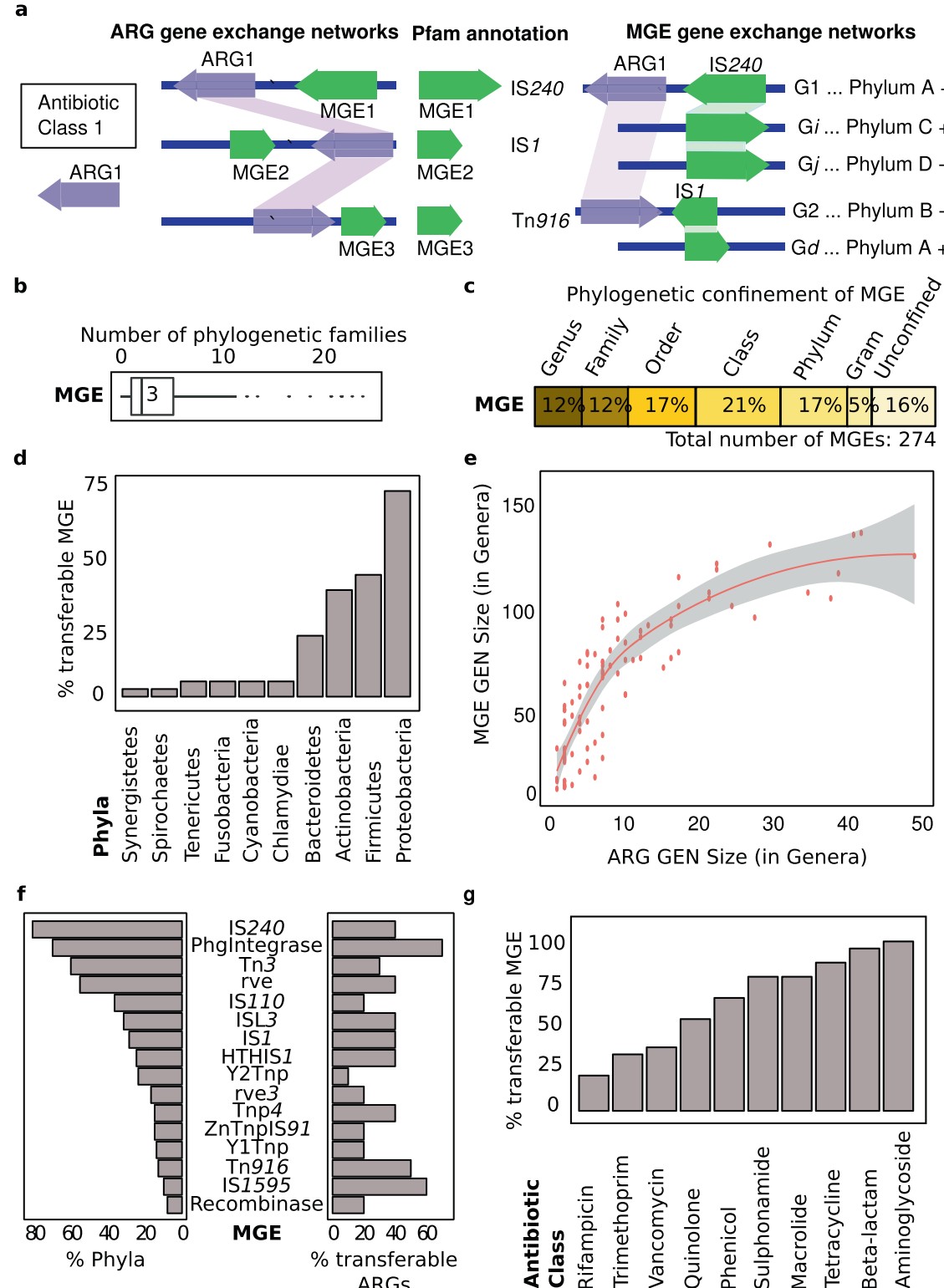

**Fig. 2 Mobile genetic elements (MGEs) in gene exchange networks (GENs). a** Conceptual guide to show how we extracted the mobile genetic elements from the ARG neighboring regions, annotated them using Pfam, and identified the MGE gene exchange network. **b** Distribution of MGEs based on phylogenetic families that participated in dissemination. Box, range of number of families; line and number, median; dots, outliers. **c** Distribution of phylogenetic confinement of MGEs. Percents of MGEs confined to a certain phylogenetic classification. **d** Percents of total MGEs that were observed in indicated phyla. **e** Distribution of ARG GEN size (as number of genera) versus associated MGE GEN size. Red line, mean values; shaded area, erorr bands as mean values +/− SEM. **f** Dissemination of MGEs across microbial phyla and their association with transferable ARGs observed in their neighbourhood. To left, n = number of phyla with a given MGE over seven phyla participating in ARGs gene exchange networks. To right, n = number of transferable ARGs neighbouring a given MGE over the total number of transferable ARGs (152 ARGs). **g** Percentage of MGEs associated with ARGs by antibiotic class to which resistance is conferred. Source data are provided in Supplementary Data 1, 3, 5, and 14.

**Table 2 Families with largest sets of transferable MGEs. Source data are provided in Supplementary Data 5 and 14.**

| Family | MGE | Percentage[a] |
|---|---|---|
| Enterobacteriaceae | 176 | 64.2 |
| Yersiniaceae | 84 | 30.7 |
| Morganellaceae | 77 | 28.1 |
| Moraxellaceae | 69 | 25.2 |
| Pseudomonadaceae | 68 | 24.8 |
| Enterococcaceae | 52 | 19.0 |
| Aeromonadaceae | 42 | 15.3 |
| Erwiniaceae | 39 | 14.2 |
| Streptococcaceae | 38 | 13.9 |
| Vibrionaceae | 36 | 13.1 |
| Staphylococcaceae | 32 | 11.7 |
| Hafniaceae | 32 | 11.7 |
| Alcaligenaceae | 32 | 11.7 |

*MGEs* mobile genetic elements.
[a]Percent of MGEs belonging to the family out of the total number of transferable MGEs.

Our analysis of dissemination risks predicted transfer of 36 ARGs to Streptococcaceae, 23 to Enterobacteriaceae, and 35 to Staphylococcaceae (Supplementary Data 9). We found evidence that 3 (~8%), 8 (~35%), and 23 (~66%) ARGs have already reached Streptococcaceae, Enterobacteriaceae, and Staphylococcaceae, respectively (Fig. 5d and Supplementary Data 11). The top genes disseminated into Enterobacteriaceae confer resistance to β-lactams (*blaZ*), macrolides (*ermT*), and aminoglycosides (*aac (6')-Ie-aph(2")-Ia*) (Fig. 5e). Genes frequently observed in Staphylococcaceae confer resistance to aminoglycosides (*aph(6)-Id* and *aph(3")-Ib*), and sulphonamides (*sul* genes). Genes frequently observed in Streptococcaceae confer resistance to aminoglycosides (*ant(9)-Ia*), β-lactams (*blaZ*), and macrolides (*erm(33)*) (Fig. 5e). We did not observe any of these ARGs in these families in our initial analysis (Fig. 1 and Supplementary Fig. 2).

**Functional compatibility constrains dissemination of ARGs.** Although genes may be mobilised by MGEs that are compatible with a potential recipient species, the resistance mechanism of the transferred ARG may not be compatible with the new host's physiology[14]. For example, basing our prediction of future dissemination solely on ARG-MGE associations means that genes involved in vancomycin resistance may reach Enterobacteriaceae family members, such as *Escherichia coli* (Fig. 3a). However, because vancomycin is not effective against Enterobacteriaceae, there is no selective pressure for the transfer of vancomycin-resistance genes to Enterobacteriaceae such as *E. coli*[14]. Other genes, such as those involved in aminoglycoside resistance, are compatible with a broad range of hosts, but may not have yet reached their full dissemination potential (Fig. 3a and Supplementary Fig. 2A). We previously showed that genes encoding enzymes conferring aminoglycoside resistance are compatible with an *E. coli* host, despite no reported genomic association[14]. Because these genes are found in Gammaproteobacteria, which are closely related to *E. coli*, and within mobile contexts detected in that host, we expect that these genes will eventually reach *E. coli* if they have not already done so.

Our analysis predicted that a high number of genes can reach many more species than currently observed (Fig. 3a, Supplementary Fig. 2A, and Supplementary Data 12). Of particular interest are β-lactamases due to their wide dissemination within the clinically important Enterobacteriaceae family[21,22]. While our in silico predictions suggested

that these genes could be much further disseminated across Gram-positive species (Fig. 3a), β-lactamases are rarely detected in these bacteria[22,23] (Fig. 1 and Supplementary Fig. 2A). This underrepresentation of β-lactamases in Gram-positive genomes, despite their association with broadly compatible mobile elements, suggests that functional constraints limits their dissemination.

To test this hypothesis, we experimentally assessed the function of 84 ARGs in the Gram-positive model organism *Bacillus subtilis*. Our results showed that ARGs conferring resistance to aminoglycoside, trimethoprim, chloramphenicol, and macrolide antibiotics will most likely function when transferred to this genus. However, the majority of these functional genes have not yet been identified in Bacillus species and some, e.g., *catI*, *tetC*, *ermA*, *mphA*, *aph(3")*, and *ant(2')*, could potentially reach this genus based on their MGE associations. None of the 25 β-lactamases, most of which are functional in *E. coli* and widely present in Gram-negative organisms, conferred a resistant phenotype in *B. subtilis* (Supplementary Data 13)[14]. This result suggested that although β-lactamases are fairly unconstrained in their movement within Gram-negative bacteria, they face a strong phylogenetic barrier limiting their dissemination across Gram-positive bacteria.

**Discussion**

We identified GENs and mobilisation elements that likely mediated the mobilisation of 152 transferable ARGs, many of which are implicated in clinical antibiotic resistance. We noted that the abundance of MGEs strongly correlated with the abundance of transferred ARGs (Fig. 2e). In a previous study, analysis of mobile ARGs and their neighbouring mobilisation elements across 23,425 genomes found that phylogeny is another major variable shaping networks for resistance gene transfer[13]. Our finding was consistent with the importance of MGEs in shaping the dissemination of ARGs across different phyla. In other words, the spread of ARGs across microbial communities is constrained by their associated MGEs. Organisms with several MGEs may be more prone to acquire and transfer ARGs. We therefore predicted the dissemination of the ARGs based on the phylogenetic reach of their associated MGEs. These predictions were partially validated using an independent genome database.

Our analysis showed that 101 ARGs could be further disseminated via the MGEs already associated with these genes. We found that several transferred ARGs were associated with more than one MGE, which would increase dissemination potential. Our prediction does not take into account the functional compatibility of an ARG with a new host. This is an important limitation as we previously showed that of 200 resistance genes, 74 did not confer resistance in *E. coli*[14]. Here, we show that certain β-lactamases that are predicted to be transferable to Gram-positive hosts based on their MGE network do not confer resistance in the Gram-positive host *B. subtilis*. This physiological limitation probably constrains ARG dissemination despite phylogenetic overlap of MGEs. Nonetheless, ~36% of our predicted new transfers were within the same bacterial family, where functional ARG expression would be likely.

Despite the large number of genomes investigated, our genomic database represented only a small sample of the global bacterial diversity. In addition, the over-representation of human pathogens in sequence databases limits broad quantitative comparisons across species. In the future, a full understanding of the dissemination of resistance genes will require a systematic large-

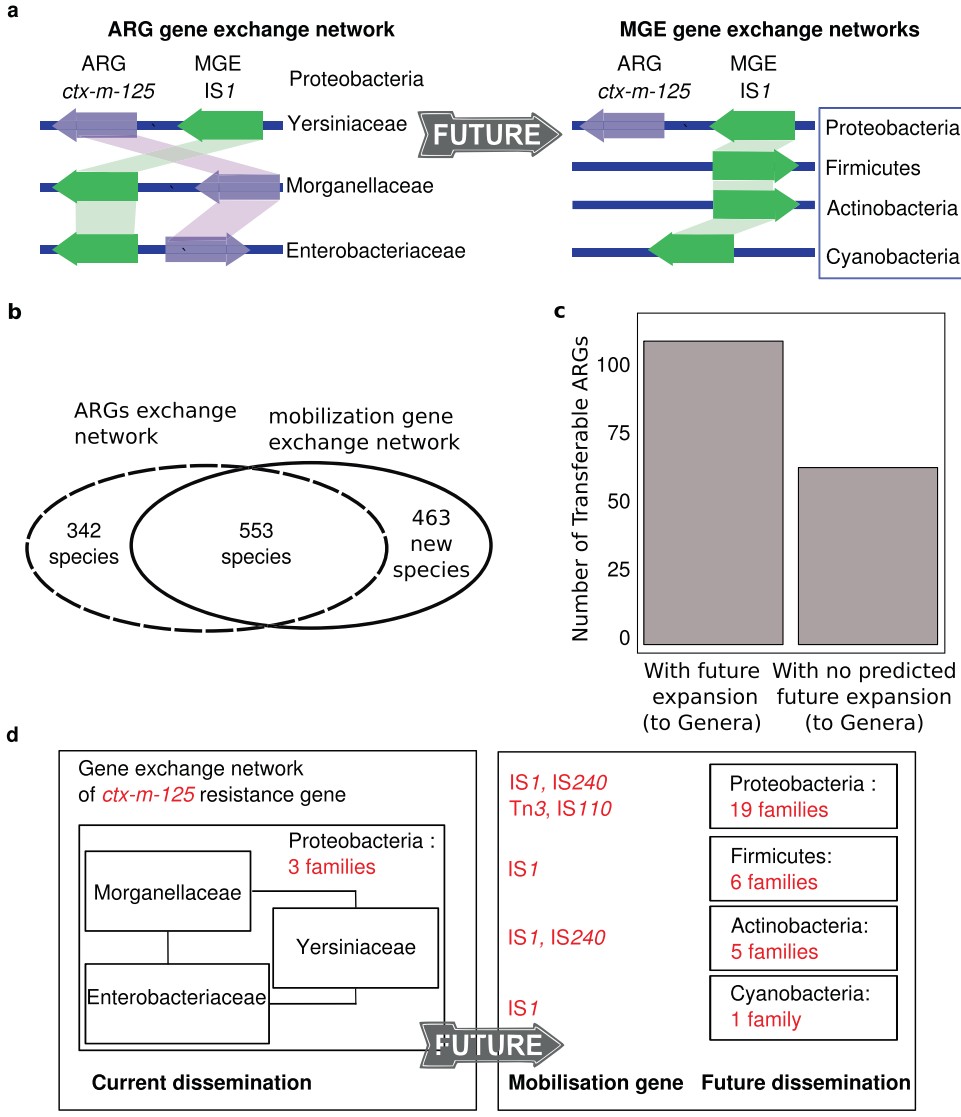

**Fig. 3 Predicted transfer of antibiotic resistance genes (ARGs) to new bacterial families. a** Conceptual guide: Left, the ARG gene exchange network, including neighbouring mobilisation contexts and phylogenetic information of the genomes. Right, ARG-associated mobile genetic element (MGE) and gene exchange network; blue-bordered boxes highlight probable phyla where ARGs may reach a new bacterial family in the future. **b** Difference and overlap of species in MGE and ARG GENs. **c** Number of transferable ARGs with potential future expansion to other genera through their MGE GEN. **d** Example: the *ctx-m-125* gene, observed in three families and expected to reach 31 new families using different MGEs. The gray arrow highlights the prediction of the *ctx-m-125* dissemination potential from its current dissemination. Boxes show future expansion of *ctx-m-125* by number of families within a phylum. Source data are provided in Supplementary Data 1, 3, 5, and 14.

scale representative sampling of the microbiome to uncover the complete extent of transfer networks, discover the potential origin species, and comprehensively plot potential future dissemination trajectories of transferable ARGs. We believe that the approach developed in this study can be the basis for ranking ARGs that pose the highest risks of antibiotic resistance dissemination. This can provide a fundamental understanding of the mechanisms and barriers to ARG dissemination to guide efforts to forecast and limit the emergence of antibiotic resistance.

## Methods

**Antibiotic resistance genes**. We compiled a comprehensive database of ARGs that includes CARD[24], ARDB[25], Jacoby-beta-lactamases[26], BacMet[27], Blad[28], CBMAR[29], resfinder[30], and our manual curation of functionally selected ARGs. The obtained sequences clustered at 98% identity and coverage using cd-hit[31].

Noisy sequences were filtered out using RESFAM[32]. To further classify ARGs, we clustered them at 95%.

**Compiled 16S rRNA**. We created a single database from these 16S rRNA databases: 16S rRNA from NCBI[33], Greengenes[34], and RDP[35]. We excluded duplicate sequences and sequences of less than 1400 bp.

**Genome database preparation, discovering and discarding contaminated genomes**. We downloaded RefSeq genomes[36] and extracted 16S rRNA gene sequences. Genomes with nearly full 16S rRNA genes of at least 1400 bp were considered in analyses. The 16S rRNA genes from all genomes were clustered at 97% identity using cd-hit[12]. Although clustering at 97% will underestimate the horizontal gene transfer between closer species, our aim was to predict the future dissemination of ARGs across phylogenetically distant strains. If 16S rRNA genes from a genome belonged to a single gene cluster, they were regarded as contamination-free genomes considered for further investigation. To ensure that genomes were properly classified, we clustered 16S rRNA genes

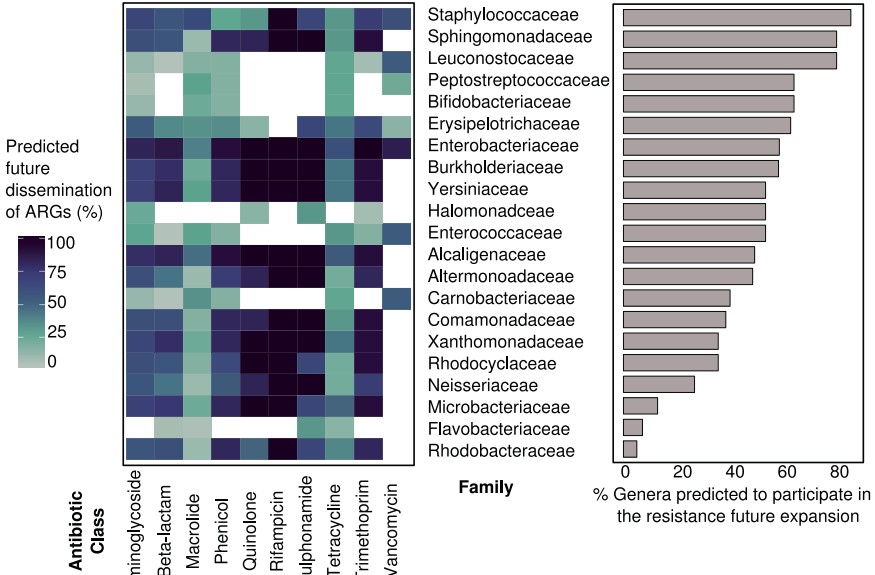

**Fig. 4 Predicted transfer of antibiotic resistance genes (ARGs) to new bacterial families.** Heatmap shows percents of transferable ARGs by affected antibiotic class with potential for future dissemination to indicated bacterial families. Barchart shows percents of genera within each family that may receive new ARGs based on their current association with a relevant mobile genetic element. Source data are provided in Supplementary Data 1, 3, 5, 10, and 14.

for all genomes in a certain species. If the 16S rRNA genes were in a single cluster, the species was considered phylogenetically consistent. Otherwise, we took the cluster with largest set of genomes as representative of the species and excluded misclassified genomes to avoid phylogenetic inaccuracy in our analyses. The final set was 56,716 genomes (Supplementary Data 14). A matrix of precomputed 16S rRNA distances was built using EMBOSS Matcher[37] as a pairwise local alignment tool for each pair of genomes from two species. Distance was calculated as

$$d = 1 - (i/(a + g)) \tag{1}$$

where $i$ is the number of identical matches, $a$ is the alignment length, and $g$ is the gaps.

**Gene exchange network prediction pipeline.** We used BLASTN[38] to compare the ARG or MGE sequences against microbial genomes. We filtered the results based on BLAST quality at 95% coverage and identity. We then extracted hit sequences including 500 bp from each side for pairwise alignment using EMBOSS Matcher. To ensure the quality of GENs, networks were filtered at 95% coverage and identity. Hit sequences were aligned using Matcher and distances calculated as in Eq. 1. Pairwise distances between 16S rRNA genes of corresponding genomes participating in a GEN were also calculated as in Eq. 1. Species were clustered allowing a maximum of 3% mismatch. GENs were statistically confirmed by comparing alignment distances between identified ARGs and 16S rRNA genes of the same genomes. P-values were calculated to determine if genes were exchanged within the network. To calculate p-values, we created two vectors, $u_{AR}$ and $u_{16srRNA}$, each with the result of the pairwise alignment of the ARG and the 16SrRNA gene for genomes participating in the GEN. The vectors included only pairwise alignment of genes from different species.

Left-tailed hypothesis testing was as follows:

$$H_0 : u_{AR} = u_{16srRNA}, \\ H_1 : u_{AR} < u_{16srRNA}, \tag{2}$$

where $u_{AR}$ is the mean distance of the hit gene distance vector and $u_{16srRNA}$ the mean distance of the 16s rRNA gene distance vector.

Rejection of the null hypothesis meant that the hit gene was more conserved than the corresponding 16s rRNA gene. Hypothesis testing was performed using the Mann–Whitney–Wilcoxon test in R (wilcox.test) with p-value of 1E−10.

**Extraction of mobile gene clusters.** We extracted 5000 bp upstream and downstream of observed transferable ARGs. Considering mobilisation elements within such close proximity of ARGs increases the likelihood of these mobilisation elements to capture the ARGs during the process of transfer. We then extracted open

reading frames (ORFs) using GeneMarkS[39]. ORFs were clustered at 95% identity and coverage using cd-hit. Clustered ORFs were annotated using COGs[40] and PFAM[41]. To identify MGEs, we applied a text mining approach[12] to COGs and PFAM annotation. We defined sequences as MGEs if text mining by both database annotation methods agreed. We filtered the MGE database to transposases, integrases, and recombinases.

**Mapping ARGs and MGEs to SRA genomes.** Whole single-cell genomes from SRA were downloaded from the National Center for Biotechnology Information (NCBI) FTP server[42]. Reads were extracted and mapped to ARGs using BLASTN. Genes were considered found only if reads mapped with a minimum of 50 bp at 95% identity and with 95% of the ARG covered by reads. SRA genomes with both MGEs and ARGs were assembled to determine if they shared a common contig. Assembly was performed using single-cell assembler SPAdes[43].

**Experimental testing of ARGs in *B. subtilis*.** ARGs (Supplementary Data 13) were selected to represent diverse mechanisms of antibiotic resistance from ARDB, CARD, and Lahey Clinic β-lactamase databases. Individual genes were ordered as gBlocks from Integrated DNA Technologies[14]. Each gene was cloned downstream of a pVeg promoter in the pDG1662 *amyE* integration vector and the chloramphenicol resistance gene of the pDG1662 vector was exchanged for the *Sh ble* Zeocin resistance gene[44]. Each construct was validated for functionality in *E. coli* and transformed into *B. subtilis* SCK6. To perform transformation, the *B. subtilis* SCK6 strain was grown in LB medium with 1 μg/ml erythromycin. The cells were cultivated at 37 °C with shaking at 300 r.p.m. overnight. The culture was then diluted to an OD 600 nm of 1.0 in fresh LB medium containing 1% (w/v) xylose to induce competence and then grown for 2 h. 0.1 μg/ml plasmid DNA was then added and the mix was incubated for 90 min[45]. Transformed cells were than selected on LB medium containing 50 μg/ml Zeocin. Antibiotic susceptibility testing was done by inoculating each antibiotic (amoxicillin, cefotaxime, mecillinam, aztreonam, meropenem, D-cycloserine, amikacin, gentamicin, trimethoprim, tetracycline, chloramphenicol, and erythromycin) starting from the wildtype minimum inhibitory concentration (MIC) and in a gradient of 2×, 5×, 10×, and 30× the wildtype MIC. Due to the resistance of *B. subtilis* SCK6 to erythromycin, the susceptible *B. subtilis* 168 was used to assess macrolide-resistance genes. Three replicate 96-well plates with 150 μl MHB2 medium (Sigma) were inoculated with $5 \times 10^5$ cells and incubated for 18 h at 37 °C with shaking at 250 rpm (Titramax 1000, Heidolph). Endpoint optical density was measured at 600 nm (Synergy H1, BioTek), and the MIC was defined as the highest concentration with lower or similar absorbance to the *B. subtilis* SCK6 (negative control) subjected to the same antibiotic concentration.

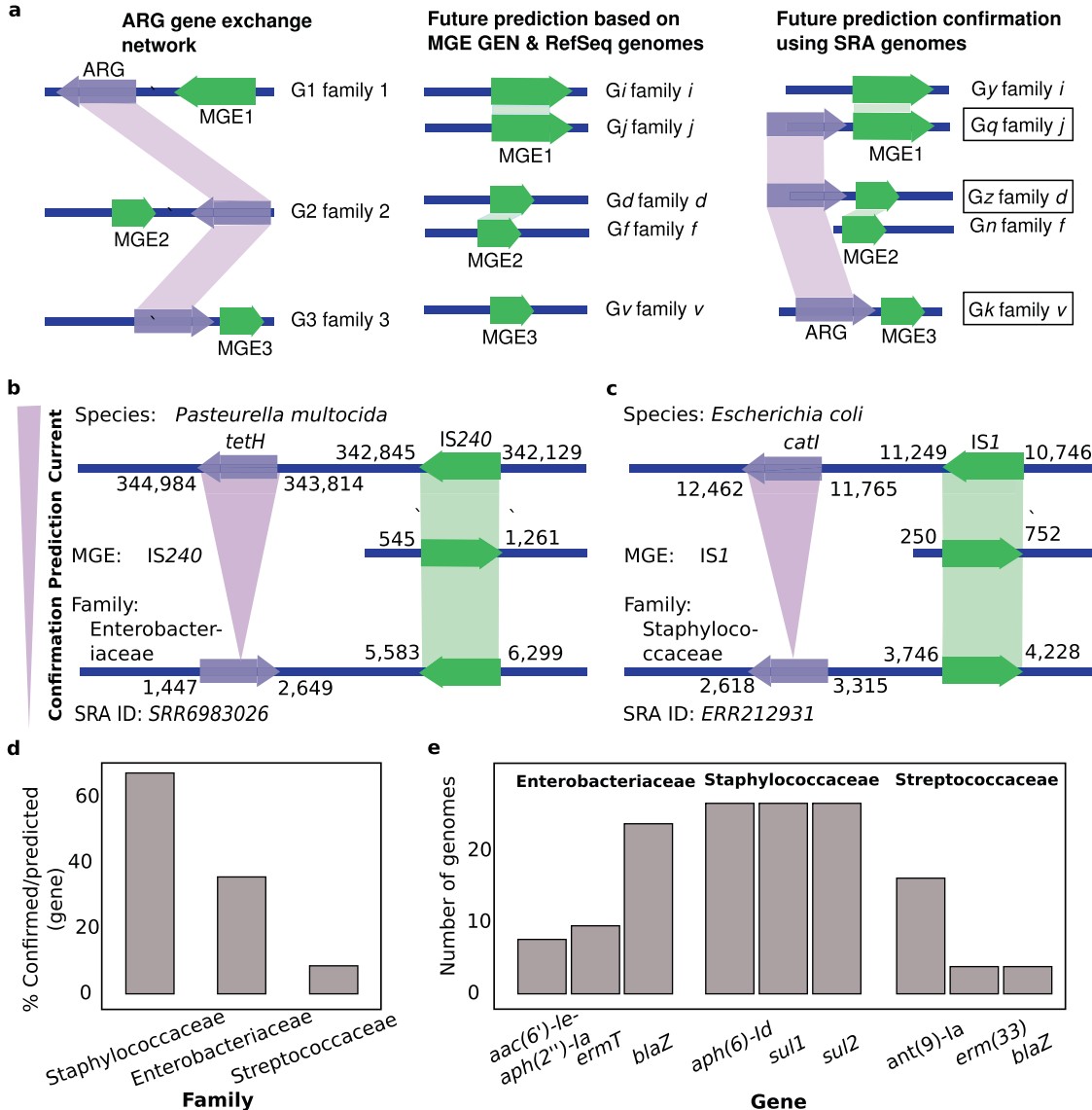

**Fig. 5 Computational confirmation analyses. a** Conceptual guide linking ARGs from gene exchange networks (GENs) to future hosts via their neighbouring MGEs and MGE GENs and confirming horizontal gene transfer predictions via SRA genomes. Boxes show confirmed transfers of ARGs to predicted hosts. **b** Example: computational confirmation for the *tetH* gene predicted in Enterobacteriaceae. The gene was observed near IS240 in Pasteurella with IS240 observed in Enterobacteriaceae genome *GCF 000693615.1* and confirmed in the Enterobacteriaceae National Centre for Biotechnology Information Sequence Read Archive (SRA) genome *SRR6983026*. **c** Example: computational confirmation using *catI* and IS1, observed in Escherichia in the current GEN with IS1 seen in Staphylococcaceae genome *GCF 000159555.1* and *catI* and IS1 found together in Staphylococcaceae SRA genome *ERR212931*. **d** Percents of predicted genes confirmed for indicated families. **e** Top three genes for indicated families that were observed to be in a significant number of genomes. Source data are provided in Supplementary Data 11.

**Reporting summary**. Further information on research design is available in the Nature Research Reporting Summary linked to this article.

## Data availability

All raw data used in this study are available in NCBI RefSeq and SRA data. We retrieved the data associated with whole-genome-sequenced bacterial genomes from both RefSeq and SRA. The numerical data underlying all figures are available in Supplementary Data 1–14. All relevant data are available from the corresponding authors.

## Code availability

Codes for pipelines for GEN, genome assembly, and annotations are available on GitHub at https://github.com/MostafaEllabaan/GEN2019 and currently available with https://doi.org/10.5281/zenodo.4553319[46].

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

## Acknowledgements

We acknowledge support from The Novo Nordisk Foundation under the NFF grant number: NNF10CC1016517. We also acknowledge support from The Lundbeck Foundation under grant agreement R140-2013-13496. We also acknowledge support from Computerome–the Danish supercomputer for life sciences—as computational power and assistance. ME would like to thank Dr. Peter Rugbjerg for his comments on the figures and the manuscripts.

## Author contributions

M.M.H.E., C.M., L.I., and M.O.A.S. designed the study program. M.M.H.E. wrote the manuscript. C.M., L.I., and A.P. designed the experimental part of the study. A.P. did the experimental part of the study and wrote the corresponding section. M.M.H.E. wrote computational pipelines and deployed and executed them for computational analysis on HPC and cloud facilities available at Computerome. All authors contributed to the editing and final revision of the manuscript.

## Competing interests

The authors declare no competing interests.
