## [Peer Review File · Nature Communications]

REVIEWER COMMENTS

Reviewer #1 (Remarks to the Author):

This manuscript analyzed over 20k bacterial genomes to gain understanding of the dissemination of antibiotic resistance genes. The study had an important and interesting question in mind. Unfortunately, the manuscript is piled with a lot of data, but the big picture is not presented well. To improve the manuscript, the authors should consider restructure the manuscript to present the data to better convey the message/significance. If the authors do that, it can also improve the readability of the manuscript. Before all these are improved, I cannot recommend it for publication.

Other comments:

Lines 38-40, "Initial mobilisation is often facilitated by transposon capture of the resistance genes followed by horizontal transfer .." Transposon capture is just one type of the mechanisms of HGT. I am not sure how well this statement holds.

Lines 65-67, "we used a statistical test based on the assumption that gene transferred horizontally between two organisms is significantly more conserved than their 16S rRNA gene." Randomly sampled gene pairs need to be examined to establish the power and sensitivity of the test. There are slow evolving genes, whether and how many vertically inherited genes can be inferred as HGT by chance.

Lines 91-93, "Even though genome databases are biased towards human pathogens, this result supports the hypothesis that the substantial selective pressure imposed on human pathogens drives the dissemination of ARGs". I am not keen to buy this argument. At a minimum, readers need to see additional information on whether or not the skewed database can explain the bias.

Lines 111-112, "Accordingly, a majority of MGEs likely have a narrow within-species host range that cannot be resolved using our method or were not sufficiently captured in our genome dataset." I would agree with the authors on this. This is also consistent with the notion of very high HGT rates among closely related strains/species. One way to partially address this may be to compare the intraspecific (within-species) diversity among different MGEs/ARGs and address the extent of transfer in different gene groups..

In the section of "Mobile genetic context predicts dissemination potential of ARGs". The authors made a potentially very important statement "the transferable resistome still has substantial dissemination potential".. Then, "..84 transferable ARGs were predicted to be able to reach a new phylum".. Since sampling among different phyla can vary remarkably, it becomes important to address whether these are in a few well-sampled phyla..

The statement "we noted that the abundance of MGEs strongly correlated with the abundance of transferred ARGs". If true, this could be interesting. In order to say this, data need to be presented to demonstrate such a correlation.

Reviewer #2 (Remarks to the Author):

This is an interesting and important study with the goal of understanding the predictive features of antibiotic resistance gene (ARG) dissemination by means of mobile genetic elements (MGE). Mobilization into new species, including potential human pathogens, is shown to be unpredictable and promiscuous, even crossing phylum barriers. The authors also rightly show the limitations of their computational approach (their prediction does not take into account the functional compatibility of an ARG with a new host, i.e. physiological constraints, which they illustrate by direct experiments in *B. subtilis*, see below).

A large collection of curated bacterial genomes from RefSeq database was used (56,716) to screen for ARGs and MGEs and a much larger collection from SRA to predict ARGs dissemination. They also experimentally validated the in-silico predictions and found that ARGs conferring resistance to aminoglycoside, trimethoprim, chloramphenicol and macrolide will most likely function (confer resistant phenotype) in Gram-positive model of *B. subtilis* when transferred from *E. coli*, while β -lactamases, most of which are functional in *E. coli* will not function in *B. subtilis*.

Major comments

Clustering at 97% 16S identity lumps together different species - thus HGTs that are reported here are an underestimate since transfer between closer species is not detected. This is by no means a weakness but should be acknowledged.

The authors excluded plasmid and phage genes "that may be involved in the function of plasmids or phages but not directly contribute to mobilisation of the ARG-containing DNA segments" - this is extremely vague and the relevant methods section does not make it clearer. If an ARG is on a plasmid that is mobile, because that plasmid has mobilization genes, then this ARG is MGE-associated and mobilizable.

"Only 274 MGEs, representing 29 MGE families, were predicted with high confidence to be transferable across species, highlighting the conservative assumptions underlying our statistical framework." Indeed this is probably an overly-conservative assumption since it is based on the 5000 bp neighbourhood and probably represents a very large under-estimate of the true mobility and hence I suggest it be removed from the manuscript, otherwise the analysis has to be extended, in the very least to whole plasmids not just 5Kb away from the ARG. Alternatively, the authors could regard this a small (but high-confidence) sample and treat it as such in the discussion.

Minor comments

Summary section (abstract):

Lines 19-20: "... and showed experimentally that physiological constraints host can explain why specific genes ..." should be "... physiological constraints of the host"

Something is grammatically wrong with the sentence, the "host" seems out of place maybe "of the host"?

Results section:

"Enterobacteria" should not be in italics, either change to the formal family name or keep the informal designation in normal type. Similarly "enterococci" should also not be italics. Phyla names such as Proteobacteria also should not be in italics.

Lines 65 - 67: This assumption that genes transferred horizontally between two organisms is should be significantly more similar than their respective 16S rRNA genes is valid for relatively recently transferred genes only. Also if this methodology has been used in previous work please cite it.

"MGEs are widely disseminated across bacterial genomes" - should be "Resistance-associated MGEs are widely disseminated across bacterial genomes"

Lines 77-80 and Table1 - why were the most highly disseminated genes ranked by the number of genera participating in their GEN (gene exchange network) and not by the actual number of species? The authors already have this information, and it might be more informative than ranking by genera.

Line 122 "phylogenetic families" - should be "bacterial families", or just "families"

Line 171-172: "Sequence Read Archive (SRA) is a comprehensive database of sequenced genomes" - this is not accurate. SRA is a public repository of sequence data (of any kind, including many raw metagenomes samples).

Line 172: the authors wrote that many of the genomes they obtained from SRA were not used in their model. They have used 472,798 genomes from SRA to test their predictions of ARGs future dissemination. Exactly how many of these 472,798 genomes on which they have tested the model were also included in the model training, if any?

Line 205 - "there is no selective pressure for the transfer of vancomycin-resistance genes to" should be "there is no selective pressure for the retention of transferred vancomycin-resistance genes to"
Line 212 "we expect that these genes will eventually reach E. coli." should be "we expect that these genes will eventually reach a strain of E. coli, if they have not already done so."

Line 231 "they face a strong phylogenetic barrier" - should be "they face a strong physiological barrier" or "... functional barrier", since it is functional incompatibility that is discussed here. Such enzymes that are generally periplasmic may be functional in G- phyla that are as remote from Proteobacteria as Firmicutes are. I am pretty sure that the corresponding author has shown this to be the case in a classic paper over a decade ago (Science 325: 1128-1131)

Why did the authors choose to focus on the families Enterobacteriaceae, Staphylococcaceae, and Streptococcaceae s to test their ARGs transfer prediction model? Important human pathogens? Then what about Enterococcaceae? The authors should clarify this.

Discussion section

The discussion section, is rather short and does not refer to previous work. Surely there have been previous studies associating ARGs with neighboring (in genomic context) MGEs? Maybe they should restructure the paper so as to combine the results with the discussion and add a short conclusion paragraph.

Methods section:

Line 272 - "green gene [34]" should be spelled as Greengenes.

Extraction of Mobile Gene Clusters subsection - why specifically was a 5,000 bp region used? upstream/downstream of ARGs was used to identify putative MGE within the neighboring genetic regions of these ARGs?

Code availability subsection - the GitHub page is incomplete, README.md is empty. The authors wish that the approach they have developed in this study will be of use to other researchers for ranking ARGs that pose a high risk to be transferred and disseminated into new species. They should add organized and detailed instructions on how to use their program/scripts. At the moment it's just a bunch of scripts with very minimal info. Furthermore, GeneExchangeNetworkPipeline folder that supposes to contain the main gene exchange network (GEN) pipeline is empty.

Figures:

In general, all the fonts in the figures should be increased.

In Figure 2F - the overlap plot (supposed to be like Venn diagram) is not clear, how many MGEs and ARGs overlap and how many do not between species?

REVIEWER COMMENTS FOR MANUSCRIPT NCOMMS-20-18969-T

Reviewer #1 (Remarks to the Author):

Comment R#1-1: This manuscript analyzed over 20k bacterial genomes to gain understanding of the dissemination of antibiotic resistance genes. The study had an important and interesting question in mind. Unfortunately, the manuscript is piled with a lot of data, but the big picture is not presented well. To improve the manuscript, the authors should consider restructure the manuscript to present the data to better convey the message/significance. If the authors do that, it can also improve the readability of the manuscript. Before all these are improved, I cannot recommend it for publication.

Response: Thanks to the reviewer comments on the readability of the manuscript. We have made major modifications to the manuscript, including re-organizing and improving the clarity of the text and of the main and supplementary figures. Specific changes are highlighted below:

1. **Section “Transferable ARGs define diverse gene exchange networks of bacteria.”** We have shortened the text (lines: 59 - 86) and improved the figures. Specifically, Figure 1 has a new panel to show how the antibiotic resistance gene and class are linked to phylogenetic class and Gram staining. The genomes of Gram-positives and Gram-negatives are integrated into Figure 1C rather than Supplementary Figure 2B. The previous Figure 1C was moved to Figure 1F for better organization of the figure and the text. Supplementary Figure 2B now has the ARG GEN distribution based genome size at different phylogenetic levels, including genera, families, and phyla, previously available in Extended Figure 1A-C.
2. **Section “Resistance-associated MGEs are widely disseminated across bacterial genomes.”** (This is now lines: 88 - 127). We have changed the figures to include a guide figure panel (Figure 2A) to show how we extracted the mobile genetic elements from the ARG neighboring regions, annotated them using Pfam, and identified the MGE gene exchange network. The figure is reorganized to better illustrate the results. The text has also been substantially reorganized. Figure 2E has been moved from the extended Figure 2A.
3. **Section “Mobile genetic context predicts dissemination potential of ARGs.”** (This is now lines: 128 - 165). We added Figure 3A to show how we used the difference between ARG GENs and MGE GENs as a tool to predict the future dissemination of ARGs. Figure 2F has been moved from Figure 3B to show the overlap and differences between the species of ARG GENs and MGE GENs. Figure 3C was previously extended Figure 2B, showing the numbers of ARGs predicted to have future expansion and otherwise. Figure 3D was Figure 3B, which shows an example of *ctx-m-125* future dissemination. Figure 4 represents Figure 3A and shows the families that will be a future host of ARGs. The text of this section has been changed and reorganized to align with the figures.
4. **Section “Predicted dissemination confirmed in Sequence Read Archive.”** (This is now lines: 166 – 202). We added Figure 5A as a conceptual guide to future predictions from both ARG GENs and MGE GENs and confirmation of ARGs dissemination using SRA. We moved

the examples to Figure 5B-C. Figure 5D-E shows the general statistics of genes confirmed in the families and the genes with the highest number of confirmed genomes in the considered families. Such reorganization, as we believe, better represents the result. The text of this section is also reorganized to better align with the current figure organization.

Other comments:

Comment R#1-2: Lines 38-40, “Initial mobilisation is often facilitated by transposon capture of the resistance genes followed by horizontal transfer ..” Transposon capture is just one type of the mechanisms of HGT. I am not sure how well this statement holds.

Response: We thank the reviewer for pointing this out. To clarify our statement, we changed the text (Lines 34-37) to “Mobile genetic elements (MGEs) such as transposons and genes that encode enzymes that facilitate them such as integrases or recombinases often facilitate the initial mobilisation. Such MGEs are capable of capturing ARGs from chromosomes and horizontally transferring them via a plasmid or a phage to other bacteria [7-11].”

Comment R#1-3: Lines 65-67, “we used a statistical test based on the assumption that gene transferred horizontally between two organisms is significantly more conserved than their 16S rRNA gene.” Randomly sampled gene pairs need be examined to establish the power and sensitivity of the test. There are slow evolving genes, whether and how many vertically inherited genes can be inferred as HGT by chance.

Response: We focus on the most recent transfer with a conservative 95% identity cutoff at low phylogenetic resolution to avoid false-positive ancient transfers. We included references to previous studies [6,7,12,13]. The text (Lines 60-62) has been changed to: “To identify putative horizontally transferred ARGs, we used a statistical test based on the assumption that genes transferred horizontally between two organisms are significantly more conserved than their 16S rRNA genes [6,7,12,13].”

Comment R#1-4: Lines 91-93, “Even though genome databases are biased towards human pathogens, this result supports the hypothesis that the substantial selective pressure imposed on human pathogens drives the dissemination of ARGs”. I am not keen to buy this argument. At a minimum, readers need to see additional information on whether or not the skewed database can explain the bias.

Response: We agree with the reviewer. Previously, we aimed to highlight the role of human activity in disseminating resistance to pinpoint that the extreme usage of antibiotics in the last 50 year led to a

higher level of disseminating resistance. However, in the revised manuscript, we focus on the future dissemination of resistance. We found that this specific statement was not well-aligned with the paper theme and topic and have removed it from the manuscript.

Comment R#1-5: Lines 111-112, “Accordingly, a majority of MGEs likely have a narrow within-species host range that cannot be resolved using our method or were not sufficiently captured in our genome dataset.” I would agree with the authors on this. This is also consistent with the notion of very high HGT rates among closely related strains/species. One way to partially address this may be to compare the intraspecific (within-species) diversity among different MGEs/ARGs and address the extent of transfer in different gene groups.

Response: While the suggestion is interesting, it was outside of the scope of our current study. Thus, unfortunately, our method is not able to observe the horizontal gene transfer within-species.

Comment R#1-6: In the section of “Mobile genetic context predicts dissemination potential of ARGs”. The authors made a potentially very important statement “the transferable resistome still has substantial dissemination potential”. Then, “.84 transferable ARGs were predicted to be able to reach a new phylum”. Since sampling among different phyla can vary remarkably, it becomes important to address whether these are in a few well-sampled phyla.

Response: At the lower phylogenetic level of genus, 101 of 151 genes will reach a new genus. Most of those events are observed in well-sampled phyla such as Proteobacteria, Firmicutes, Actinobacteria and Bacteroidetes. Note that we did not consider functional compatibility in this analysis. In addition, beta-lactamase genes may not be successful in Gram-positive bacteria despite predictions otherwise. We observe that the transferable resistome still has substantial dissemination potential as shown in Figure 3C and Supplementary Figure 4. Text has been updated on Lines 139 - 148.

Comment R#1-7: The statement “we noted that the abundance of MGEs strongly correlated with the abundance of transferred ARGs”. If true, this could be interesting. In order to say this, data need to be presented to demonstrate such a correlation.

Response: We agree with the reviewer and refer to Figure 2E, which shows the distribution of ARG gene exchange network (GEN) sizes (as number of genera) versus associated MGE GEN size. We changed the text in lines 242-244 from "we noted that the abundance of MGEs strongly correlated with the abundance of transferred ARGs" to "we noted that the abundance of MGEs strongly correlated with the abundance of transferred ARGs (Figure 2E)."

We thank the reviewer for the comment to improve the readability of the manuscript.

Reviewer #2 (Remarks to the Author):

This is an interesting and important study with the goal of understanding the predictive features of antibiotic resistance gene (ARG) dissemination by means of mobile genetic elements (MGE). Mobilization into new species, including potential human pathogens, is shown to be unpredictable and promiscuous, even crossing phylum barriers. The authors also rightly show the limitations of their computational approach (their prediction does not take into account the functional compatibility of an ARG with a new host, i.e. physiological constraints, which they illustrate by direct experiments in *B. subtilis*, see below).

A large collection of curated bacterial genomes from RefSeq database was used (56,716) to screen for ARGs and MGEs and a much larger collection from SRA to predict ARGs dissemination. They also experimentally validated the *in-silico* predictions and found that ARGs conferring resistance to aminoglycoside, trimethoprim, chloramphenicol and macrolide will most likely function (confer resistant phenotype) in Gram-positive model of *B. subtilis* when transferred from *E. coli*, while β -lactamases, most of which are functional in *E. coli* will not function in *B. subtilis*.

Response: Thanks for the positive comments. We greatly appreciate them.

Major comments

Comment R#2-1: Clustering at 97% 16S identity lumps together different species - thus HGTs that are reported here are an underestimate since transfer between closer species is not detected. This is by no means a weakness but should be acknowledged.

Response: Line 287-290 notes that 97% clustering has been considered by several studies to distinguish species at higher resolution [12]. We agree with the reviewer that clustering at 97% will underestimate horizontal gene transfer between closer species. Although horizontal gene transfer is more likely among closely related species, our aim was to predict the future dissemination of ARGs across phylogenetically distant strains. We now state this in lines 107-109: "Accordingly, a majority of MGEs likely have a narrow within-species host range that cannot be resolved using our method or were not sufficiently captured in our genome dataset."

Comment R#2-2: The authors excluded plasmid and phage genes "that may be involved in the function of plasmids or phages but not directly contribute to mobilisation of the ARG-containing DNA segments" – this is extremely vague and the relevant methods section does not make it clearer. If an

ARG is on a plasmid that is mobile, because that plasmid has mobilization genes, then this ARG is MGE-associated and mobilizable.

Response: We thank the reviewer for pointing this out. In the revised manuscript we include an explanation of why we excluded plasmid or phage elements that may not contribute to transfer of ARGs (lines 94 - 101). We included only genes associated with indicator genes implicated in conjugation. It is hard to know from contigs alone if a plasmid is mobilisable etc...

Comment R#2-3: “Only 274 MGEs, representing 29 MGE families, were predicted with high confidence to be transferable across species, highlighting the conservative assumptions underlying our statistical framework. “ Indeed this is probably an overly-conservative assumption since it is based on the 5000 bp neighbourhood and probably represents a very large under-estimate of the true mobility and hence I suggest it be removed from the manuscript, otherwise the analysis has to be extended, in the very least to whole plasmids not just 5Kb away from the ARG. Alternatively, the authors could regard this a small (but high-confidence) sample and treat it as such in the discussion.

Response: We agree with the reviewer. We regard MGEs within 5000 bp of a neighbouring region of ARGs as a small (but high-confidence) sample. We updated the manuscript lines 100-101 as follows “e have adopted this very conservative approach to avoid false positives. We considered the resulting pool of MGEs as a small but high-confidence sample (**Supplementary Table 4**).”

Minor comments:

Comment R#2-4: Summary section (abstract): Lines 19-20: "... and showed experimentally that physiological constraints host can explain why specific genes ..." should be "... physiological constraints of the host" Something is grammatically wrong with the sentence, the "host" seems out of place maybe "of the host"?

Response: We changed the lines 17-18: "... and showed experimentally that physiological constraints host can explain why specific genes ..." to "... and showed experimentally that physiological constraints of the host can explain why specific genes ...”

Comment R#2-5: “Enterobacteria” should not be in italics, either change to the formal family name or keep the informal designation in normal type. Similarly “enterococci “ should also not be italics. Phyla names such as Proteobacteria also should not be in italics.

Response: Thanks for this comment. The genera and phyla names were changed to nonitalic form across the manuscript including tables and figures.

Comment R#2-6: Lines 65 - 67: This assumption that genes transferred horizontally between two organisms should be significantly more similar than their respective 16S rRNA genes is valid for relatively recently transferred genes only. Also, if this methodology has been used in previous work please cite it.

Response: We are interested in the most recent transfers that capture the impact of humans since the discovery of antibiotics and the impact of the heavy antibiotic use in clinics, farming and agriculture. This hypothesis has been considered in several studies including [6,7,12,13]. Line 60-62 has been updated to include these references. In addition, we use the current dissemination pattern to predict the future dissemination of ARGs. Recent transfers provide better tools for predicting the future. Considering ancient transfers would increase the chance of false positives as the mobilisation elements may be reorganised away from the neighborhood of ARGs around them. We acknowledged this in lines 107-109, "Accordingly, a majority of MGEs likely have a narrow within-species host range that cannot be resolved using our method or were not sufficiently captured in our genome dataset."

Comment R#2-7: "MGEs are widely disseminated across bacterial genomes" - should be "Resistance-associated MGEs are widely disseminated across bacterial genomes"

Response: Line 89 was changed from "MGEs are widely disseminated across bacterial genomes" to "Resistance-associated MGEs are widely disseminated across bacterial genomes."

Comment R#2-8: Lines 77-80 and Table1 - why were the most highly disseminated genes ranked by the number of genera participating in their GEN (gene exchange network) and not by the actual number of species? The authors already have this information, and it might be more informative than ranking by genera.

Response: The number of genera shows the diversity of ARGs hosts better than the species. Therefore, we kept the genera and ranking as is. Note that our methods clustered the genomes at 97% 16SrRNA identity. Such clustering may lead to a collapse of close species together, making genus a more reliable phylogenetic level than the species to measure ARG dissemination.

Comment R#2-9: Line 122 "phylogenetic families" - should be "bacterial families", or just "families".

Response: Thanks for the comments to improve the readability of the article. Line 111, "phylogenetic families" changed to "bacterial families."

Comment R#2-10: Line 171-172: "Sequence Read Archive (SRA) is a comprehensive database of sequenced genomes" - this is not accurate. SRA is a public repository of sequence data (of any kind, including many raw metagenomes samples).

Response: We changed line 168-170 from "Sequence Read Archive (SRA) is a comprehensive database of sequenced genomes" to "SRA is a public repository of sequence data (of any kind, including many raw metagenomes samples). Here, we only considered the whole bacterial genome sequenced data in SRA."

Comment R#2-11: Line 172 (now line 170): the authors wrote that many of the genomes they obtained from SRA were not used in their model. They have used 472,798 genomes from SRA to test their predictions of ARGs future dissemination. Exactly how many of these 472,798 genomes on which they have tested the model were also included in the model training, if any?

Response: In our future prediction confirmation, we excluded the ARGs observed in the family. If the ARG was observed previously in the family, the family was not considered a future host. If the RefSeq genomes overlapped with SRA genomes, the entire family that the genomes belonged to was not counted because it already had the ARG. Thus, the entire family was not counted as a future host of this ARG.

Comment R#2-12: Line 212 (now lines 217-218)“we expect that these genes will eventually reach *E. coli*.” should be “we expect that these genes will eventually reach a strain of *E. coli*, if they have not already done so.”

Response:, We changed the text from “we expect that these genes will eventually reach *E.coli*” to “we expect that these genes will eventually reach a strain of *E. coli* if they have not already done so” (lines: 217-218).

Comment R#2-13: Why did the authors choose to focus on the families Enterobacteriaceae, Staphylococcaceae, and Streptococcaceae to test their ARGs transfer prediction model? Important human pathogens? Then what about Enterococcaceae? The authors should clarify this.

Response: For Sequence Read Archive analysis, we wanted to confirm our prediction using genomic data available in SRA. We chose these three families because at the time of download, they represented more than 50% of the SRA bacterial genomes (~887,000 genomes) in the the deposited data. These families represent both Gram-positive and Gram-negative bacteria. They are the most sampled families. They are also the most common human pathogens. In addition, they fulfilled the purpose of the study. In lines: 172-175, we added the following statement to address the reviewer comment: “The three families were chosen because of their deposited data and represent more than 50% of the SRA bacteria genomes (~ 887,000 genomes). These families represent both Gram-positive and Gram-negative bacteria, are the most sampled families, and include many common human bacterial pathogens.”

Discussion section

Comment R#2-14: The discussion section is rather short and does not refer to previous work. Surely there have been previous studies associating ARGs with neighboring (in genomic context) MGEs? Maybe they should restructure the paper so as to combine the results with the discussion and add a short conclusion paragraph.

Response: We have restructured the paper significantly, as detailed in the Reviewer R#1-1 comment. We compared our results to previous methods as detailed in lines 244-247, and the concluding paragraph has also been changed in Lines 271-274. With the significant restructuring of the manuscript, the paper's readability has significantly improved. We believe having a separate section for the results and discussion makes the paper well-organized and presents the results well—although this depends upon the reviewer's approval.

Methods section:

Comment R#2-15: Line 272 - "green gene [34]" should be spelled as Greengenes.

Response: We changed line 283: "green gene [34]" to "Greengenes [34]"

Comment R#2-16: Extraction of Mobile Gene Clusters subsection - why specifically was a 5,000 bp region used? upstream/downstream of ARGs was used to identify putative MGE within the neighboring genetic regions of these ARGs?

Response: We updated lines 325-327 to justify using 5000 bp. The following statement has been added: “Considering mobilisation elements within such close proximity of ARGs increases the likelihood of these mobilisation elements to capture the ARGs during the process of transfer. ”

Comment R#2-17: Code availability subsection - the GitHub page is incomplete, README.md is empty. The authors wish that the approach they have developed in this study will be of use to other researchers for ranking ARGs that pose a high risk to be transferred and disseminated into new species. They should add organized and detailed instructions on how to use their program/scripts. At the moment it's just a bunch of scripts with very minimal info. Furthermore, GeneExchangeNetworkPipeline folder that supposes to contain the main gene exchange network (GEN) pipeline is empty.

Response: Thank you for the comments, we updated GitHub. We also included a docker image on the Dropbox to allow reproduction of our results. Please note that comprehensive reproduction of this result requires extensive computational resources due to the massive data used. Therefore we recommend running analyses on high-performance computing facilities or cloud platforms.

Comment R#2-18: In general, all the fonts in the figures should be increased.

Response: Font size was increased for all figures.

Comment R#2-19: In Figure 2F - the overlap plot (supposed to be like Venn diagram) is not clear, how many MGEs and ARGs overlap and how many do not between species?

Response: Thank you for the comment. The Venn diagram was corrected and showed how many MGEs and ARGs overlapped and how many did not between species in Figure 3B, which was previously Figure 2F.

REVIEWERS' COMMENTS

Reviewer #1 (Remarks to the Author):

The readability of the manuscript still needs to be improved. For the introduction, I would rather like to see less details but more preparation for the readers in the introduction. The results section has the meat. I still feel that the study is tackling a very important question and the scale of the analysis is good. The GEN pipeline is the core of their analysis. The authors may want to introduce/mention the GEN pipeline earlier in the manuscript in a way that readers can get and also appreciate the importance and novelty of the analysis. Overall, the current version of the manuscript is still very heavy on data piling. If the authors can build the significance of the study and then provide the data, the paper will be able to reach more audience.

Reviewer #2 (Remarks to the Author):

All my comments have been adequately addressed by the authors.

REVIEWER COMMENTS FOR MANUSCRIPT NCOMMS-20-18969A

Reviewer #1 (Remarks to the Author):

The readability of the manuscript still needs to be improved. For the introduction, I would rather like to see less details but more preparation for the readers in the introduction. The results section has the meat. I still feel that the study is tackling a very important question and the scale of the analysis is good. The GEN pipeline is the core of their analysis. The authors may want to introduce/mention the GEN pipeline earlier in the manuscript in a way that readers can get and also appreciate the importance and novelty of the analysis. Overall, the current version of the manuscript is still very heavy on data piling. If the authors can build the significance of the study and then provide the data, the paper will be able to reach more audience.

Response:

We have updated our manuscript, especially the introduction section. The introduction section has been changed accordingly, discussing all the previous work briefly first, followed by a concise summary of the current work. The significance and importance of the gene exchange network pipeline have also been highlighted in the introduction sections: 78-94. We hope that the current version of the manuscript has shown the importance and significance of the current study and help reach more audiences. Thanks for the reviewer comments that help us improve the manuscript's quality, allowing it to reach more audiences.

Reviewer #2 (Remarks to the Author):

All my comments have been adequately addressed by the authors.

Response:

We thank all reviewers and editors for their help and support throughout the revision.